

# Evaluation of the effect of chickenpox vaccination on shingles epidemiology using agent-based modeling

Ellen Rafferty[1], Wade McDonald[2], Weicheng Qian[2], Nathaniel D. Osgood[2] and Alexander Doroshenko[3]

[1] School of Public Health, University of Saskatchewan, Saskatoon, SK, Canada
[2] Department of Computer Science, University of Saskatchewan, Saskatoon, SK, Canada
[3] Faculty of Medicine and Dentistry, Department of Medicine, Division of Preventive Medicine, University of Alberta, Edmonton, AB, Canada

Corresponding author
Alexander Doroshenko,
adoroshe@ualberta.ca

## ABSTRACT

**Background:** Biological interactions between varicella (chickenpox) and herpes zoster (shingles), two diseases caused by the varicella zoster virus (VZV), continue to be debated including the potential effect on shingles cases following the introduction of universal childhood chickenpox vaccination programs. We investigated how chickenpox vaccination in Alberta impacts the incidence and age-distribution of shingles over 75 years post-vaccination, taking into consideration a variety of plausible theories of waning and boosting of immunity.

**Methods:** We developed an agent-based model representing VZV disease, transmission, vaccination states and coverage, waning and boosting of immunity in a stylized geographic area, utilizing a distance-based network. We derived parameters from literature, including modeling, epidemiological, and immunology studies. We calibrated our model to the age-specific incidence of shingles and chickenpox prior to vaccination to derive optimal combinations of duration of boosting (DoB) and waning of immunity. We conducted paired simulations with and without implementing chickenpox vaccination. We computed the count and cumulative incidence rate of shingles cases at 10, 25, 50, and 75 years intervals, following introduction of vaccination, and compared the difference between runs with vaccination and without vaccination using the Mann–Whitney U-test to determine statistical significance. We carried out sensitivity analyses by increasing and lowering vaccination coverage and removing biological effect of boosting.

**Results:** Chickenpox vaccination led to a decrease in chickenpox cases. The cumulative incidence of chickenpox had dropped from 1,254 cases per 100,000 person-years pre chickenpox vaccination to 193 cases per 100,000 person-years 10 years after the vaccine implementation. We observed an increase in the all-ages shingles cumulative incidence at 10 and 25 years post chickenpox vaccination and mixed cumulative incidence change at 50 and 75 years post-vaccination. The magnitude of change was sensitive to DoB and ranged from an increase of 22–100 per 100,000 person-years at 10 years post-vaccination for two and seven years of boosting respectively ($p < 0.001$). At 75 years post-vaccination, cumulative incidence ranged from a decline of 70 to an increase of 71 per 100,000 person-years for two and seven years of boosting respectively ($p < 0.001$). Sensitivity analyses had a minimal impact on our inferences except for removing the effect of boosting.

**Discussion:** Our model demonstrates that over the longer time period, there will be a reduction in shingles incidence driven by the depletion of the source of shingles reactivation; however in the short to medium term some age cohorts may experience an increase in shingles incidence. Our model offers a platform to further explore the relationship between chickenpox and shingles, including analyzing the impact of different chickenpox vaccination schedules and cost-effectiveness studies.

## INTRODUCTION

Varicella (chickenpox) and herpes zoster (shingles) are two diseases caused by the varicella zoster virus (VZV). Individuals are generally infected with chickenpox in childhood. In Canada prior to vaccination, approximately 11.7 per 1,000 persons were infected with chickenpox each year, which was estimated to cost about 122 million CAD$ annually (*National Advisory Committee on Immunization, 2010*). Following primary chickenpox infection, the VZV migrates to the sensory nerve ganglia, where it remains latent and can subsequently reactivate as shingles, which causes a dermatomal rash often accompanied by itching and pain (*Cohen, 2013*). Shingles was estimated to occur at a rate of 1.2–3.4 per 1,000 per year (3.9–11.8 cases per 1,000 per year in those >65 years) in Canada before chickenpox and shingles vaccination (*Public Health Agency of Canada, 2016*). Some studies suggest that the incidence of shingles has been increasing in the recent decades prior to the introduction of chickenpox vaccination (*Russell et al., 2014*; *Hales et al., 2013*; *Ogunjimi, Van Damme & Beutels, 2013*).

Although much remains unknown about shingles and the reactivation process, research has demonstrated that the maintenance of latency is largely governed by VZV cell-mediated immunity (VZV-CMI). Successful reactivation of shingles occurs when VZV-CMI weakens to a certain (unknown) level, which is often a result of the normal ageing process (*Levin, 2012*). However, the literature shows there are likely other risk factors that can result in inadequate immune response to VZV reactivation at any age, including mental health, stress, comorbid infections (e.g., cytomegalovirus (CMV) infection) and therapy, and disease-related immunosuppression (e.g., HIV-AIDS) (*Levin, 2012*). In parallel to the natural waning of VZV-CMI immunity, one theory posits that exogenous boosting of VZV-CMI is a determinant of shingles reactivation (*Hope-Simpson, 1965*; *Garnett & Grenfell, 1992*; *Garnett & Ferguson, 1996*). Exogenous boosting occurs when a VZV immune individual is exposed to a case of chickenpox or shingles.

The theory of exogenous VZV-CMI boosting has sparked a debate about whether the chickenpox vaccine will limit the boosting of immunity to shingles and therefore increase the incidence of the disease, however the empirical data to-date is largely inconclusive (*Russell et al., 2014*; *World Health Organization, 2014*). This debate has prompted many countries, including the United Kingdom and France, to delay the implementation of a universal chickenpox vaccine, even though there is a safe and effective vaccine available

(*European Centre for Disease Prevention and Control, 2015*). In comparison, other countries, including Canada and the USA, currently recommend a two-dose chickenpox vaccination schedule (*Public Health Agency of Canada, 2010*). Alberta introduced a universal one-dose chickenpox vaccination program for 12-month-olds in 2002 and in 2012 added a second dose for children aged 4–6 years. *Russell et al. (2014)* found that shingles incidence was increasing in Alberta in the period 1994–2010, both before and after the introduction of the chickenpox vaccination.

The unique connection between chickenpox and shingles makes independently exploring their disease dynamics difficult. Using mathematical modeling to study the interacting factors related to these diseases—in particular, how chickenpox vaccination may impact shingles disease and epidemiology—complements conventional epidemiological studies and offers a unique approach to evaluating the effects of vaccination in a controlled study using simulated data.

Most previous modeling studies of shingles and chickenpox have been conducted using aggregate population-level models (*Ogunjimi, Van Damme & Beutels, 2013*). Both agent-based and aggregate models can simulate the indirect effects of varicella vaccination, including the changing risk of disease over time, herd immunity, age-category specific mixing and rates, and increasing age of infection. However, agent-based models (ABMs) provide a number of advantages in the study of many infectious diseases, including allowing the exploration and measurement of disease dynamics at multiple levels and the adjustment of both individual and population parameters (*Ahmed, Greensmith & Aickelin, 2012*; *Marshall et al., 2015*; *Osgood, 2007*; *Osgood, 2009a*). In the study of the VZV, ABMs can provide for realistic and comprehensive simulation of the transmission of infection and boosting of immunity by explicit modeling of network-mediated contacts, and flexibility in specifying interpersonal contact. ABMs allow the simulation of between-host dynamics (e.g., transmission) at an individual level, while also capturing spatial limits on such transmission and allowing the simulation of continuous within-host dynamics, including for both aging and individual waning of VZV-CMI; the characterization of such factors as continuous processes offers flexibility in terms of reporting and characterization of individual-level dynamics (*Marshall et al., 2015*; *Osgood, 2004*, *2007*; *Osgood 2009b*). ABMs can further represent detailed elements of vaccination, including vaccine attitudes, uncertainties in vaccine coverage and—critically for this study—continuous waning of vaccine immunity. Importantly for supporting options for later expansions and refinements of this model, the individual-level representation can readily depict vaccination attitude dynamics based on family context. As ABMs support reporting an individual's disease or vaccination status in light of their history, catch-up immunization and breakthrough illness can also be represented in a modular manner that scales well to more complex vaccination regimes. Finally, ABMs take into account stochastics and readily accommodate large number of dimensions of heterogeneity, including sex, age, spatial location, comorbid conditions, and detailed information on condition-specific individual history (*Osgood, 2004*, *2007*; *Osgood, 2009a*, *2009b*).

A systematic review of empirical and modeling studies on the boosting of VZV-CMI for shingles found that while there is evidence that boosting of VZV-CMI exists, there is little evidence of the strength or duration of that boosting effect (*Ogunjimi, Van Damme & Beutels, 2013*). The majority of chickenpox vaccination models that incorporated boosting have predicted there will be an increase in shingles incidence post chickenpox vaccination; however most of them have assumed a high force of boosting (*Ouwens et al., 2015*; *Brisson et al., 2010*; *Guzzetta et al., 2016*). While modeling studies have explored the plausibility of various types of boosting (e.g., progressive immunity, partial immunity, temporary immunity) (*Guzzetta et al., 2016*, *2013*) there is little exploration of the duration of boosting (DoB) and the rate at which an individual's immunity to VZV wanes over time, and how changing assumptions regarding these factors may impact the rate of shingles following chickenpox vaccination.

Within this work, we sought to develop an ABM of chickenpox and shingles disease and vaccination based on current immunological, medical and epidemiological data, and replicate the epidemiology of chickenpox and shingles in Alberta, Canada before chickenpox vaccination. In Alberta, universal chickenpox vaccine has been in place since 2002, and detailed population demographics and chickenpox and shingles epidemiological data both before and after vaccine implementation were available. The primary objective of this study was to determine how chickenpox vaccination in Alberta impacts the incidence and age-distribution of shingles over 75 years post-vaccination taking into consideration a variety of plausible quantitative theories of waning and boosting of VZV immunity.

## METHODS

We developed an ABM using the multi-method simulation software AnyLogic® Professional (version 7.3.7), that represents chickenpox transmission, chickenpox and shingles disease and vaccination states, as well as the waning and boosting of VZV immunity (https://figshare.com/articles/Chickenpox_and_shingles_ABM/5294647/1). The model time unit is one year, however, the model operates in continuous time, where an event may occur at any arbitrary point in time, driven by either hazard rates (e.g., births, exposures), or by occurrence of other events (e.g., due to receipt of an "exposure" message sent by another agent) or at fixed time after occurrence of another event. The model is initialized after a 75-year burn-in period, with any calibration or experimentation taking place after that period. We included such a burn-in period representing the average lifespan, to ensure that individuals were at different stages of waning of VZV immunity and therefore at different risk of getting shingles. This study was approved by the Health Ethics Research Board at the University of Alberta (study ID Pro00068334).

### Model structure and agent-characteristics

Statecharts related to within-host dynamics of the model are shown in Fig. 1A. A disease statechart where agents are in protected, susceptible or disease states determined an agent's probability of contracting chickenpox or shingles. The chickenpox and shingles vaccination schedules' statecharts represented which vaccination each agent received.

The chickenpox vaccination schedule was modeled based on the Alberta VZV vaccination policy (*Government of Alberta, 2015*). Agents become due for chickenpox vaccines at the ages of 12 months for the first dose and at 4–6 years—for the second dose (Fig. 1B). A representation of one-dose shingles vaccination given to individuals 50 years or older is depicted in the statechart shown in Fig. 1C; however, currently shingles is not part of the publicly funded schedule in Alberta and therefore was not included as part of this analysis. An agent receives a vaccine dose with a probability based on its vaccine attitude as described below. If an agent receives the second dose but has not yet received the first, it may also receive a catch-up of the first dose with a certain probability. These probabilities are specified by parameters, as shown in Table 1. A demographic statechart represented Alberta births, ageing and mortality characteristics (Fig. 1D).

An agent's chance of being infected with chickenpox was dependent on whether they came into contact with someone with infection and the risk of transmission on such a contact. In comparison, an agent's likelihood of getting shingles was determined by their individual waning immunity timer (i.e., after chickenpox infection, a countdown on "Immunity Waning Time" initiates; when it completes, the agents become susceptible to shingles) as well as the number and degree of boosting they received. The immunity waning time was represented by a formula derived from *Ogunjimi et al. (2015)* and further by calibration (Fig. S1; Table 2). This equation inherits assumptions whereby force of reactivation is represented by a gamma distribution, initial CMI is represented by a normal distribution and the waning of immunity rate for shingles is a fixed rate that can be altered using a coefficient—waning of immunity coefficient (WoI) (*Ogunjimi et al., 2015*). Furthermore, to account for the small but sharp increase in childhood shingles cases as observed in *Russell et al. (2014)*, we included a small proportion of the population between the ages of 0 and 19 (5%) who had a short waning of immunity timer, allowing the model to account for individuals who may have a weaker force of boosting, lower initial CMI (e.g., immunocompromised) or a very quick waning of the immunity rate.

An agent's chance of getting shingles was determined by the number of years they were protected through boosting. The model assumed progressive boosting, as postulated by *Guzzetta et al. (2016)* and therefore the number of years of boosting protection was calculated by multiplying the number of times an agent comes into significant contact with the VZV by the duration in years of each boosting event. The number of boosting events was determined through the distance-based contact network and the duration of each boost was equal to the number of years of added protection from shingles derived through each boost. Altering the quantitative values of DoB and WoI of shingles resulted in significant changes in the incidence of shingles in a population.

## Contacts, network and spatial context

The VZV model represented agents in a stylized geographic area, where agents are connected to other agents based on their proximity to one another (a distance-based network) (Fig. S2). When agents are randomly placed in the model environment they are connected to all other agents within their connection range (*connectionRange_Norm*). We included a low-density periphery and a higher density central region to better
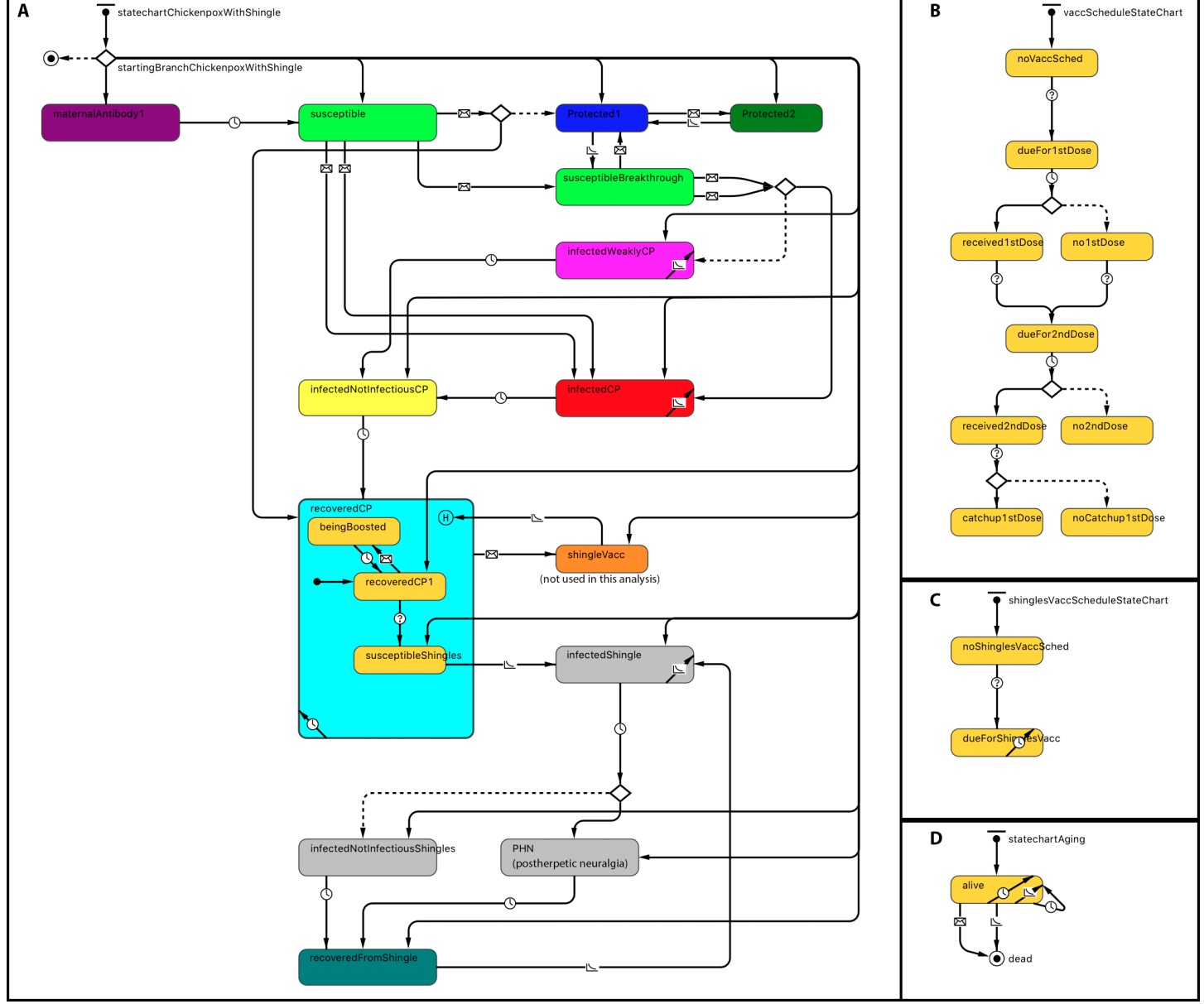

**Figure 1 Statechart structure.** (A) Disease and protection. (B) Chickenpox vaccination schedule. (C) Shingles vaccination schedule. (D) Demographics.

represent how disease spreads in a typical public health district in Alberta (or comparable jurisdiction) spanning an urban center and rural regions. Approximately 20% of our population were part of low density regions and the remainder, 80% resided in a high-density region.

The *populationDensityUrban* and the *populationDensityRural* determine the number of agents in a respective area per unit of distance in the model and along with the *connectionRange* determine the number of connections between agents. The contact rate (*baseContactRate_Norm*) determines the number of contacts (e.g., messages) an agent

**Table 1 Data sources, key parameter values and values for calibration.**

| Parameter category | Parameter name | Description | Value | Reference or calibration |
|---|---|---|---|---|
| Demographics | Population size (Persons) | Population size at the model's initialization. | 500,000 | |
| | Mortality and fertility rates | Life tables for Alberta and pregnancy outcomes (live birth) for Canada by age group were used to estimate mortality rates and fertility rates in our population | Data available online | *Statistics Canada (2008, 2016a, 2016b)* |
| Disease Mechanisms | Initial cell-mediated immunity VZV | The distribution across the population of cell-mediated immunity for shingles derived from initial infection with varicella zoster virus. (Fig. S1) | Max (0.001, normal (0.05,1)) | *Ogunjimi et al. (2015)* |
| | Force of reactivation | Represents the distribution of force of reactivation for shingles. It is a unitless value that is compared to the initial cell-mediated immunity to calculate the 'waning of immunity time'. (Fig. S1) | Gamma distribution (2,0.1,0) | *Ogunjimi et al. (2015)* |
| | Waning of immunity coefficient shingles (WoI) | Coefficient to determine the annual loss of protection based on VZV-CMI. | Values tested in calibration: 0.45–0.93<br><br>Values included in the analysis: 0.50–0.74 | Calibration-3[§] |
| | Waning of immunity rate shingles | Annual loss of protection based on VZV-CMI. | 0.4 | *Ogunjimi et al. (2015)* |
| | Duration of exogenous boosting (DoB) | Number of years before protection returns to previous levels following a boost. | Values tested in calibration: 0.42–10<br><br>Values included in the analysis: 2–7 | Calibration-3[§] |
| Disease Propagation | Exogenous infection rate (1/Year) | Represents rate per year of chickenpox infection imported from outside the model population. | 17.83 | Calibration-2[*] |
| | Probability of infection on contact message | Represents the probability of infection per contact *message* received by a susceptible agent in a model. | Normal: 0.78<br>Breakthrough: 0.234<br>Shingles: 0.234 | Calibration-1[¶]<br>*Gershon, Takahashi & Seward (2012)* |
| Network characteristics | Connection Range (Length) | Distance of an individual's connection range. The range depends on whether agents were included in the preferential mixing age or the normal mixing age. | Preferential Range = 21.245<br><br><br><br><br>Normal Range = 8.958 | Calibration-1[¶];<br>*Mossong et al. (2008)* |

(Continued)

| Parameter category | Parameter name | Description | Value | Reference or calibration |
|---|---|---|---|---|
| | Base contact rate (1/Day) | Number of contacts per agent per day, which dependent on if agents were part of the preferential or normal age range. | Preferential contact rate = 20; Normal contact rate = 30.124 | Calibration-1[¶]; *Mossong et al. (2008)* |
| | Shingles connection range modifier | A ratio to lower the connection range of individuals with HZ to make it less infectious than CP. | 0.124 | Calibration-2[*] |
| | Preferential mixing age (Year) | Age group where we have increased the connection range and base contact rate to better reflect the dynamics in the population. | 1–9 years | *Kwong et al. (2008)* |
| | Population density (Agents per length) | Represents the number of agents per arbitrary distance for urban and rural population. This parameter in combination with connection range determines the number of connections an agent has in our model. | Urban: 0.3 Rural: 0.2 | |
| Chickenpox vaccine parameters | Vaccination attitude in the population (%) | Distribution of vaccine rejecters, hesitant and acceptors in the population. | Acceptor = 65, Hesitant = 30, Rejecter = 5 | Vaccine coverage generated by the model was referenced by vaccine coverage reported by *Alberta Health (2017)* |
| | Probability Catch-Up (%) | Probability that an individual will get a catch-up vaccine when due for second dose vaccination. | 55 | |
| | Probability first dose vaccination (%) | Probability an individual will get first dose vaccination given their vaccine attitude. | Acceptor = 97, Hesitant = 75, Rejecter = 3 | |
| | Probability second dose vaccination (%) | Probability an individual will get second dose vaccination given they received first dose vaccine. | Acceptor = 98, Hesitant = 82, Rejecter = 33 | |
| | Primary vaccine failure chickenpox (%) | The percent of individuals that do not have an immune response to CP vaccination. | 1st dose = 16–24 2nd dose = 5–16 | *Gershon, Takahashi & Seward (2012) Bonanni et al. (2013) Duncan et al. (2017)* |
| | Waning of chickenpox vaccine immunity (1/Year) | The rate that chickenpox vaccine immunity wanes each year. | 1st dose protected = 0.02 2nd dose protected = 0.00 | *Gershon, Takahashi & Seward (2012)* |

**Notes:**

[¶] Calibration-1: Contact and mixing patterns as well as the probability of infection per contact message were determined by varying input parameters by hand until the model outputs approximated age-specific incidence of chickenpox, adjusted for under-reporting.

[*] Calibration-2: Automated (algorithmic) AnyLogic calibration experiment to determine the values of exogenous infection rate and shingles connection range modifier.

[§] Calibration-3: Testing a range of plausible values for DoB and WoI by statistically fitting to the empirical incidence rates for chickenpox and shingles.

**Table 2  Calibration results.**

| | Duration of Boosting-DoB (years) | Waning of Immunity Coefficient- WoI (1/year) | P-values[1] |
|---|---|---|---|
| Combination1[2] | 0.42 | 0.45 | <0.001 |
| Combination 2 | 2 | 0.50 | 0.051 |
| Combination 3 | 3 | 0.55 | 0.313 |
| Combination 4 | 4 | 0.60 | 0.052 |
| Combination 5 (Baseline Scenario) | 5 | 0.63 | Reference |
| Combination 6 | 6 | 0.68 | 0.963 |
| Combination 7 | 7 | 0.74 | 0.121 |
| Combination 8 | 8 | 0.79 | 0.001 |
| Combination 9 | 9 | 0.85 | <0.001 |
| Combination 10 | 10 | 0.93 | <0.001 |

**Notes:**

[1] *P*-values for the Mann–Whitney U-test comparing age-specific shingles incidence sum of residuals squared for DoB 5 years to all other scenarios. Calibrations scenarios deemed statistically not different (i.e., *p*-value > 0.05) were included in the main experiment. Calibration scenarios which were statistically different from the best-fit calibration experiment were excluded.

[2] Represents combinations of DoB and WoI; all other parameters in the model stayed the same.

makes to a connected agent per day. It is through these messages, sent through connections and contacts, that individuals can "transmit" chickenpox infection and provide boosting of VZV. To represent the increased contact time and range of connectivity among day-care and early school-aged children, we incorporated different connection ranges and contact rates based on age (*Mossong et al., 2008*). Those aged one to nine years were considered "preferential contacts" in our model, and therefore have a higher contact rate (*baseContactRate_Pref*) and connection range (*connectionRange_Pref*) when interacting with individuals within the same age range. We chose this group to represent "Preferential Contacts" as these are the age groups with the highest rates of chickenpox infection as described by *Kwong et al. (2008)*. These network adjustments ensured more realistic contact network assumptions than those of random-mixing and compartmental models, such that not only age-group preferences of contacts were captured, but also increased global connectivity due to bridging effects of younger age groups was considered, both in a spatial context. To characterize the transmissibility of VZV, we included parameters whereby a susceptible agent will have a certain probability of infection per contact message received from a normal chickenpox case (*probCPDiseaseOnContact*), a breakthrough case of chickenpox (*probCPDiseaseOnContactWithBreakthrough*) and a shingles infection (*probCPDiseaseOnContactWithShingles*). The probability of infection in our model is dependent on whether a susceptible agent would come in contact with and receive a message from an infected case of chickenpox and shingles. For example, an infected agent in the preferred age group (i.e., a child) may send 20 messages per day, each to a random connected agent. Considering that the mean number of connections per agent in this age group is 10.87, the number of messages per connection-day equals to 20/10.87 = 1.8 ≈ 1–2. Outside of the preferred age group, this equates to 3/7.42 = 0.40 < 1 message per connection-day (Table 1; https://figshare.com/articles/Chickenpox_and_shingles_ABM/5294647/1).

Some of these messages do not lead to a receiving agent becoming infected because the receiving agent is not susceptible, or an agent is not selected by a model to be connected to an agent sending a message (not being part of contact pool). A complete age-specific connection and contact frequency profile is available at https://figshare.com/articles/ Chickenpox_and_shingles_ABM_connection_and_contact_frequency_profile/5552763/1.

## Parameterization

To parameterize the model, we conducted a comprehensive literature review of chickenpox and shingles disease, including modeling, epidemiological, and immunology studies and drew evidence from Alberta's Interactive Health Data Application (IHDA) (*Alberta Health, 2017*). The main parameters are listed in Table 1. We drew demographic data, including Alberta life tables, and population and age distributions, and Canadian fertility rates from *Statistics Canada (2008)*, *Statistics Canada (2016a*, *2016b)*. Chickenpox vaccination parameters, such as those associated with primary vaccine failure and waning of vaccination immunity were derived from literature (*Gershon, Takahashi & Seward, 2012*; *Bonanni et al., 2013*; *Duncan et al., 2017*). We built the mechanism whereby vaccine coverage was generated by our model based on distribution of vaccination probabilities and population vaccination attitudes as described by *Doroshenko, Qian & Osgood (2016)*. We classified all individuals into three groups: those who accept, reject and are hesitant to receive vaccination, and we assigned vaccination probabilities for each of these groups. We used calibration to ensure that model-generated vaccine coverage rates were comparable to those reported in Alberta (*Alberta Health, 2017*). For the baseline scenario in the main experiment, chickenpox vaccination maintained an average coverage for the first dose of 85.58% (95% CI [85.54–85.62]) and second dose coverage of 80.28% (95% CI [80.24–80.32]) across all years and all model runs.

## Calibration and validation of the model

We calibrated our model using a step-wise approach.

First, contact and mixing patterns, as well as the probability of infection per contact message, were determined using a pattern-oriented modeling approach. Pattern-oriented modeling involves identifying multiple patterns of behavior in a complex system and constructing model structures and parameters that replicate those patterns (*Grimm, 2005*). The rationale for using this approach is that transmission of infection in our model (as in real life) is driven by combination of the location of agents, pathogen transmissibility per discordant contact and intrinsic characteristics of agents reflecting their susceptibility (or lack of) to infection. Therefore, to represent this multi-faceted transmission in our model, we varied three model parameters *by hand* (connection range, contact rate, and probability of infection on contact message), until the model outputs approximated empirical data, specifically age-specific incidence rates for chickenpox (*Kwong et al., 2008*). Calibration for this variable was completed once our model generated patterns of disease very similar to what was observed in empirical data (Fig. 2). The best values for probability of infection on contact message, contact rates and connection ranges derived from this calibration process are shown in Table 1. The

probability of transmitting infection from a chickenpox breakthrough case and from shingles case were assumed to be 0.3 of the probability of transmitting infection from a normal chickenpox case (*Gershon, Takahashi & Seward, 2012*). As we only had empirical data for medically-attended chickenpox (*Kwong et al., 2008*), we tested different underreporting factors and found that reported chickenpox cases constituted approximately 40% of all cases in our model (an underreporting factor of 2.5, consistent with previous studies that suggest the degree of underreporting can range between 2.5 and 7.7 in a country where chickenpox is notifiable) (*Ciofi Degli Atti et al., 2002*). To further validate our model, we tested model outputs against secondary attack rates for chickenpox reported in the literature. Our model generated a secondary attack rate of 41% in comparison to attack rates observed in the household settings, ranging mostly between 60% and 78% among unvaccinated contacts (but reported as low as 31% among individuals > 15 years old) (*Gershon, Takahashi & Seward, 2012*; *Ceyhan, Tezer & Yildirim, 2009*; *Seward et al., 2004*). We considered this a realistic estimate as contacts in our population intuitively should be less intensive than the household contacts represented in the literature. Furthermore, the overall chickenpox incidence and the age-specific proportion of the population with varicella antibody when vaccination was disabled in the model was consistent with Canadian data prior to vaccination (Fig. S3) (*Public Health Agency of Canada, 2012*; *Ratnam, 2000*).

Second, we ran an automated AnyLogic calibration experiment to determine the values for four other unknown parameters in our model, DoB, WoI (which was used to calculate the waning of immunity rate), exogenous infection rate (i.e., the number of chickenpox cases brought in from outside the simulated population) and shingles connection range (i.e., modification factor for the connection range for people with shingles, to account for the closer contact required to spread VZV through shingles in comparison to chickenpox). The calibration experiment *automatically* varied these parameters over continuous ranges in order to minimize the objective function quantifying the difference (square root of the average of square differences) between the model and actual data (i.e., age-specific incidence of shingles as described in *Russell et al. (2014)*) (Fig. 2). A full description of the data used in model fitting is shown in Table S1. This automated calibration experiment predicted an exogenous infection rate of 17.8 per year and a shingles connection range modifier of 0.124. These values are shown in Table 1. Furthermore, based on the automated experiment values for DoB of 0.42 years, and a WoI of 0.45/year, we produced age-specific incidence rates for shingles and chickenpox consistent with empirical data.

Third, using the automated calibration values described above as a starting point, we then further investigated whether different combinations of DoB and WoI could also reproduce empirical data. We compared the incident rates resulting from a range of plausible values for DoB and WoI to the empirical incidence rates for chickenpox and shingles (*Russell et al., 2014*; *Kwong et al., 2008*; *Topping, Høye & Olesen, 2010*; *Railsback & Grimm, 2012*). At this stage, we considered paired values of DoB and WoI to have satisfied calibration if the resulting overall rate of chickenpox and shingles, over a model run of 100 years on 50,000 population, was within 10% of the empirical values (*Topping, Høye & Olesen, 2010*;
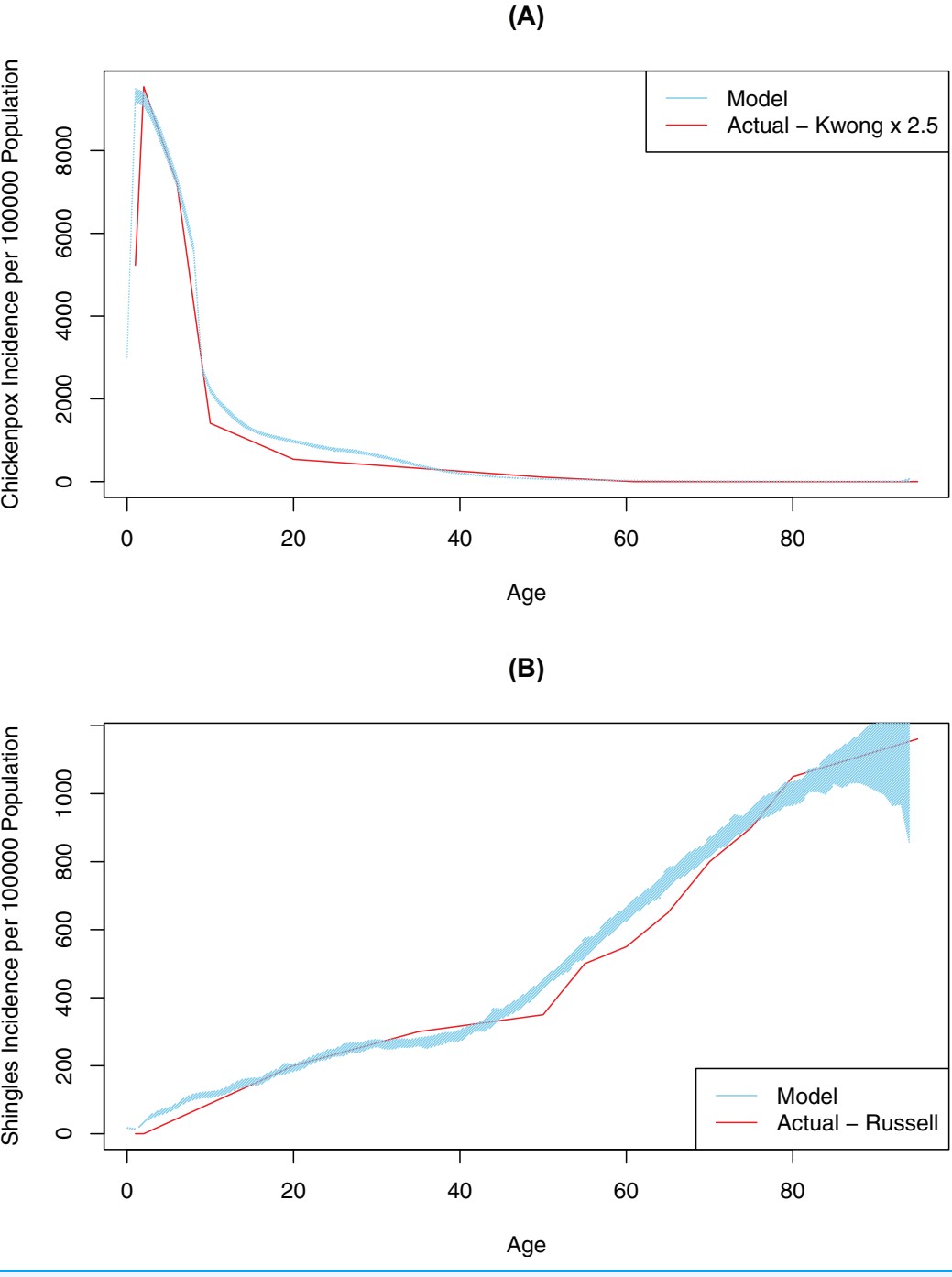

**Figure 2 Model-generated (blue line) and published (red line) age-specific incidence rates for chickenpox and shingles at time 0: model calibration—baseline scenario.** (A) Model data is based on multiple simulations for the baseline scenario; empirical data as described by *Kwong et al. (2008)*; best fit is achieved at 2.5 multiple of empirical data. (B) Model data is based on multiple simulations for the baseline scenario; empirical data as described by *Russell et al. (2014)*. In all images the blue polygon represents pointwise minimum and maximum values.

*Railsback & Grimm, 2012*). Based on these guidelines, we found nine other values for a combination of DoB and WoI that met this step of calibration.

Finally, we then ran the 10 combinations that met the criteria noted above for a period of 100 years on a 500,000-population sample (10-fold larger than in the previous step). Output from each run was then compared to empirical data on age-specific incidence rates for chickenpox and shingles. We calculated the absolute median difference between model incidence rates of shingles and empirical data for each age group to determine which combination had the smallest difference as determined by lowest sum of residuals squared. The combination with the lowest sum of residuals squared (27.95) for shingles was then considered our baseline scenario in our main experiment (which represented best fit) (https://figshare.com/articles/Sum_of_residuals_squared_-_VZV_model/5466556/1). The sum of residuals squared from each combination was then compared to the baseline scenario and tested for statistically significant difference using the Mann–Whitney U-test at the 5% level of significance. Based on the findings of the statistical tests and as a form of scenario analysis, we included both the baseline scenario along with any other DoB and WoI combinations that were statistically not different (*p*-value > 0.05) as scenarios in our main experiment.

## Main experiment

For each of the six scenarios chosen for the main experiment, we conducted at least 30 paired runs, with and without chickenpox vaccination. Running paired runs for each scenario allowed the measurement of differences in shingles rates with and without vaccination for each DoB and WoI combination. Paired model runs were given the same random seed to ensure consistency between pairs. Chickenpox vaccination was represented as part of a two-dose schedule, as described above. In our model, chickenpox vaccination started at 25 years after the initialization of the model and continued for 75 years. The count and cumulative incidence rate of shingles cases were compared at the 5% level of significance using the Mann–Whitney U-statistical test between the runs with and without chickenpox vaccination at four different time periods, specifically at 10, 25, 50, and 75 years, following introduction of vaccination.

## Sensitivity analysis

We ran several sensitivity analyses that tested how varying vaccination parameters may impact the count of shingles cases following chickenpox vaccination. Specifically, we compared a one-dose to a two-dose chickenpox vaccine schedule. In a separate set of analyses, we compared coverage rates by moving 10% of hesitant individuals to vaccine acceptors (higher vaccine coverage) in one sensitivity analysis and 10% of hesitant individuals to vaccine rejecters (lower vaccine coverage) in the second sensitivity analysis. Furthermore, we tested the impact of removing the boosting of shingles immunity (i.e., positing no added years of protection on contact with a chickenpox/shingles case) to see the overall impact of removing this biological effect from the shingles incidence estimates both before and after chickenpox vaccination.

## RESULTS

### Input calibration

Based on the last step of our calibration we found that the age-specific shingles residuals for DoB five years were not significantly different to five of the other combinations (Table 2). These combinations were then included in our main experiment as six scenarios (Scenario 2–7, sum of residuals squared for shingles ranged from 27.95 to 41.51, *p*-value > 0.05) (https://figshare.com/articles/Sum_of_residuals_squared_-_VZV_model/5466556/1). All sum of residuals squared for chickenpox by age group were not statistically different for each calibration scenarios (sum of residuals squared for chickenpox ranged from 249.20 to 251.00, *p*-value > 0.05). As one of the calibration validity tests, we tried to fit the model to empirical data with no boosting, however when we disabled boosting we could not replicate the age-specific incidence rate observed in Alberta prior to vaccination (*Russell et al., 2014*).

### Main experiment

Chickenpox vaccination led to a large drop in chickenpox cases across all six scenarios. In the baseline scenario, the cumulative incidence of chickenpox had dropped from 1,254 cases per 100,000 person-years pre-chickenpox vaccination to 193 cases per 100,000 person-years 10 years after the vaccine implementation. The cumulative incidence of chickenpox was further reduced to 49 cases per 100,000 person-years 75 years following vaccination. In comparison, all scenarios from 10-years and 25-years post-vaccination showed significantly greater shingles incidence with vaccination compared to the no-vaccination (Table 3). However, the degree of this increase and its subsequent decline was markedly different between experiments (Fig. 3; Table 3). For instance, in Scenario 2 there was an increase of approximately 22.47 cases per 100,000 person-years after 10 years; by comparison, in Scenario 7 the magnitude of the increase was greater at 99.71 cases per 100,000 person-years over that decade (*p* < 0.001). At 75 years post-vaccination, cumulative incidence ranged from a decline of 69.6 to an increase of 71.3 per 100,000 person-years for two and seven years of boosting respectively (*p* < 0.001). The increase in the incidence of shingles was positively associated with the DoB and showed a positive relationship with cumulative shingles incidence rising with each incremental increase in DoB. By the 75-year interval, the shingles incidence in all experiments was below the rate with no-vaccination (Fig. 3); however, the cumulative incidence in Scenarios 5–7 was still higher with vaccination than without. In all our scenarios, vaccination did eventually lower the overall rate of shingles; however, the amount of time was dependent on the DoB and the WoI.

At 10-year interval, all age groups greater than 10 years old showed small but equal increase in the number of shingles cases. However, at each subsequent time point (as a greater number of persons of younger age are protected with chickenpox vaccination from initial infection with the virus), cases of shingles were progressively more concentrated in the older age-groups. In contrast, in the younger age group we observed a decrease in shingles almost immediately following chickenpox vaccination (Fig. 4).

We noted several interesting observations in our model. In the no-vaccination model runs, there was an overall increase in shingles cases even without vaccination; in the baseline scenario, the incidence of shingles was 418 per 100,000 person-years after 10 years and increased to 425 per 100,000 person-year after 75 years. Moreover, the average number of boosts per individual calculated in our model over their life-time (with no chickenpox vaccination) was 1.83 with majority of people receiving 0 or 1 boosts. We found minor differences in chickenpox and shingles incidence between urban and rural communities. The incidence of shingles was slightly higher in rural settings in the absence of vaccination but became lower with vaccination (Fig. S4). Approximately 5% of urban agents had connections with rural agents (Fig. S10).

## Sensitivity analysis

None of the vaccination sensitivity analyses had a major impact on the number of shingles cases post-vaccination. As expected, we saw an increase in the number of shingles cases with a higher number of vaccine acceptors and therefore higher coverage than baseline. Conversely, we saw fewer shingles cases post chickenpox vaccination when we only gave one dose of the vaccine and when we had more vaccine rejecters, however, the effect was minimal. The difference between the low coverage (higher number of vaccine rejecters) and the high coverage (lower number of vaccine rejecters) was only 2 shingles cases per 100,000 person-years. By contrast, removing the biological effect of boosting had a substantial impact on the number of shingles cases both before and after vaccination, with cases rising to levels much higher than what was seen in Alberta (*Russell et al., 2014*). Without chickenpox vaccination, the shingles rate was very high at 734 cases per 100,000 person-years. Simultaneously, this is the one analysis where the rates begin to decline immediately following vaccination with a decline in cumulative incidence of 215 per 100,000 person-years 75 years after the implementation of the chickenpox vaccine (Table 4).

## DISCUSSION

Our ABM successfully simulated chickenpox and shingles dynamics over time by creating a 500,000-person, distance based-contact network, with detailed representation of boosting and waning of immunity. During the calibration process we re-scaled chickenpox incidence rates to account for the underreporting of chickenpox due to use of the rate of medically-attended chickenpox in published estimates (*Kwong et al., 2008*). We calibrated the model to six different scenarios for DoB and WoI for shingles, suggesting that many different quantitative values for these two unknown parameters are consistent with the empirical data. Based on this calibration, we determine that shingles incidence post-vaccination is highly sensitive to the values for both DoB and WoI, although all scenarios eventually led to a reduction in shingles incidence rates relative to baseline rates.

Infectious disease models can provide valuable insight into the complex relationship between chickenpox and shingles, allowing epidemiologists and biologists to test theories, study the impact of different parameters, and judge the outcomes of various

**Table 3** Change in all-ages cumulative incidence of shingles over 75 years after implementation of chickenpox vaccination, by scenario and time period.

| Scenario number | Time periods | | | | | | | |
|---|---|---|---|---|---|---|---|---|
| | T0–T10 | | T0–T25 | | T0–T50 | | T0–T75 | |
| | Cumulative incidence-chickenpox vaccination scenario[1] | Change in cumulative incidence[2,3] (95% CI) | Cumulative incidence-chickenpox vaccination scenario | Change in cumulative incidence (95% CI) | Cumulative incidence-chickenpox vaccination scenario | Change in cumulative incidence (95% CI) | Cumulative incidence-chickenpox vaccination scenario | Change in cumulative incidence (95% CI) |
| Scenario 2 | 405.68 | 22.47 (21.91, 22.99) | 402.88 | 18.36 (17.94, 18.78) | 375.12 | −8.64 (−9.06, −8.21) | 313.53 | −69.60 (−70.11, −69.09) |
| Scenario 3 | 452.80 | 41.13 (40.24, 42.02) | 461.01 | 48.18 (47.72, 48.65) | 437.09 | 23.92 (23.46, 24.37) | 365.03 | −48.38 (−48.96, −47.81) |
| Scenario 4 | 492.59 | 59.93 (59.02, 60.83) | 515.86 | 82.16 (81.62, 82.69) | 496.86 | 61.43 (60.88, 62.00) | 414.04 | −22.45 (−23.22, −21.67) |
| Baseline scenario | 490.00 | 71.32 (70.19, 72.45) | 530.17 | 109.23 (108.48, 109.98) | 520.47 | 97.38 (96.69, 98.09) | 434.75 | 9.57 (8.66, 10.47) |
| Scenario 6 | 513.03 | 85.33 (83.22, 87.43) | 574.61 | 145.00 (143.50, 146.50) | 572.32 | 139.06 (137.80, 140.31) | 476.03 | 40.04 (38.89, 41.20) |
| Scenario 7 | 538.11 | 99.71 (97.98, 101.44) | 627.29 | 185.73 (184.10, 186.51) | 632.65 | 185.73 (184.85, 186.61) | 522.70 | 71.33 (70.46, 72.21) |

**Notes:**

[1] Average shingles cumulative incidence with chickenpox vaccination per 100,000 person-years (averaged over 30 or more model runs).

[2] Change in shingles cumulative incidence per 100,000 person-years calculated as the average shingles incidence with chickenpox vaccination minus the average shingles incidence without chickenpox vaccination. Positive number represents an increase in cumulative incidence and negative number—a decrease.

[3] Using the Mann–Whitney U-test all changes in cumulative incidence for every time and scenario combination were statistically significant ($p < 0.05$).

interventions. To date the majority of chickenpox and shingles models, including the only model representing a Canadian population (*Brisson et al., 2000*) are aggregate compartmental models, which limit their flexibility and heterogeneity (*Ouwens et al., 2015*; *Riche et al., 2016*; *Betta et al., 2016*; *Marziano et al., 2015*; *Poletti et al., 2013*; *Gao et al., 2010*). To our knowledge, the only individual-based model was created by *Ogunjimi et al. (2015)* to combine within- and between-host dynamics, and VZV immunological data to estimate boosting characteristics. Our model utilized their representation of the biological conditions for chickenpox and shingles infection and VZV-CMI; however, it differs from the Ogunjimi model in several key ways. Ogunjimi's model represented contacts using a probabilistic method whereas ours implemented a distance-based network and included an increased contact range for school-aged children (*Ogunjimi et al., 2015, 2009*). The Ogunjimi model used a population that remained fixed in size and was based on Belgium demographics, while our model implemented an open and non-fixed population based on Alberta data, with realistic demographic changes over time. Furthermore, the Ogunjimi model assumed 100% vaccine effectiveness and incorporated only a one-dose chickenpox vaccination schedule, while ours included representations of vaccine attitudes as a dynamic predictor of vaccine coverage and probability of primary vaccine failure.

Our model concurs with previous models and biological studies that suggest exogenous boosting of VZV immunity is a likely factor in the reactivation of VZV, as

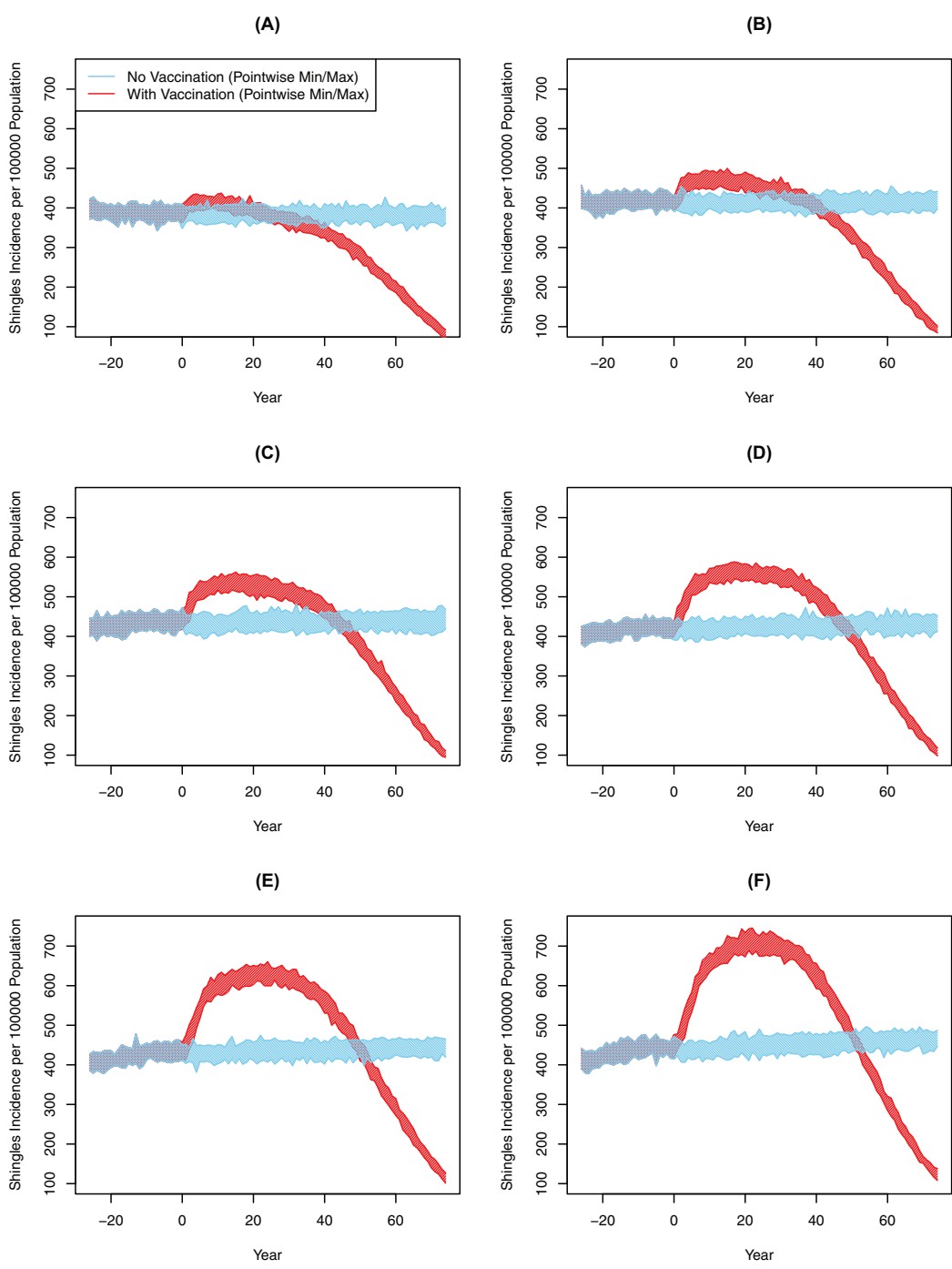

**Figure 3 All-ages shingles annual incidence over time after implementing chickenpox vaccination by duration of boosting, multiple simulations.** (A) Scenario 2 (DoB = 2). (B) Scenario 3 (DoB = 3). (C) Scenario 4 (DoB = 4). (D) Scenario 5 (DoB = 5). (E) Scenario 6 (DoB = 6). (F) Scenario 7 (DoB = 7). In all images the blue and red polygons represent pointwise minimum and maximum values.

we were not able to recreate the empirical data observed in Alberta without incorporating some element of boosting (*Ogunjimi, Van Damme & Beutels, 2013*). Longitudinal immunological studies show individuals re-exposed to chickenpox, either on a one-time

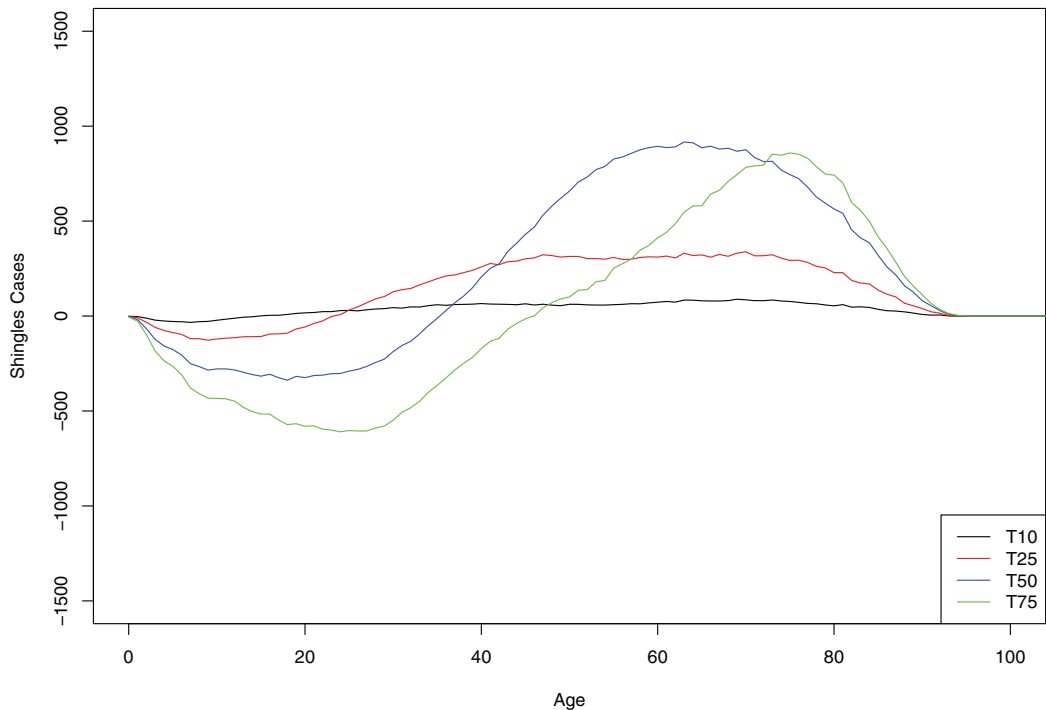

**Figure 4 Mean cumulative count of shingles cases added/averted by the age group and time point, baseline scenario.** Positive number on the *y*-axis indicates the number of shingles cases added and negative number—the number of shingles cases averted.

or continuous basis, have a corresponding increase in VZV-specific immunity (*Arvin, Koropchak & Wittek, 1983*; *Vossen et al., 2004*; *Ogunjimi et al., 2014*). However, these studies generally only look at the short-term immunological effects of boosting (up to one year post-exposure) and not everyone is boosted following re-exposure, raising questions about the DoB and the degree, number and quality of exposures needed to produce a boost of VZV-CMI (*Ogunjimi, Van Damme & Beutels, 2013*). A recent study showed that only 17–25% of grandparents who were exposed to chickenpox received a significant boost in VZV-specific immunity and that this boost typically lasted less than a year (*Ogunjimi et al., 2017*).

Using our ABM, we varied these unknown boosting parameters (e.g., DoB, WoI, degree and probability of boosting on contact) to see how those variations impacted the outcomes of chickenpox vaccination. By varying DoB and WoI simultaneously, we identified several boosting of immunity scenarios that could fit current Alberta data. In comparison, many previous chickenpox and shingles models have made some strict assumptions about boosting of immunity, with the parameter values for the force or DoB set high. For instance, *Ouwens et al. (2015)* assumed that the force of boosting would be equal to the force of infection, while *Brisson et al. (2010)* postulated each boost would result in 24 years of protection. It is perhaps not surprising that under these assumptions, many of the compartmental models predicted an increase in shingles following chickenpox vaccination.

**Table 4 Sensitivity analysis—change in all-ages cumulative incidence of shingles over 75 years after implementation of chickenpox vaccination, by scenario and time period.**

| Sensitivity analysis number[3] | Time periods | | | | | | | |
|---|---|---|---|---|---|---|---|---|
| | T0–T10 | | T0–T25 | | T0–T50 | | T0–T75 | |
| | Cumulative incidence-chickenpox vaccination scenario[1] | Change in cumulative incidence[2,4] (95% CI) | Cumulative incidence-chickenpox vaccination scenario[1] | Change in cumulative incidence (95% CI) | Cumulative incidence-chickenpox vaccination scenario | Change in cumulative incidence (95% CI) | Cumulative incidence-chickenpox vaccination scenario | Change in cumulative incidence (95% CI) |
| Baseline scenario | 490.00 | 71.32 (70.19, 72.45) | 530.17 | 109.23 (108.48, 109.98) | 520.47 | 97.38 (96.69, 98.09) | 434.75 | 9.57 (8.66, 10.47) |
| Sensitivity analysis 1 | 489.17 | 70.49 (69.30, 71.67) | 527.77 | 106.83 (106.08, 107.58) | 514.28 | 91.20 (90.50, 91.89) | 435.04 | 9.86 (9.01, 10.71) |
| Sensitivity analysis 2 | 488.77 | 70.09 (68.88, 71.31) | 529.12 | 108.18 (107.39, 108.97) | 519.36 | 96.28 (95.64, 96.91) | 434.68 | 8.02 (8.70, 10.30) |
| Sensitivity analysis 3 | 490.54 | 71.86 (70.48, 73.24) | 530.73 | 109.79 (109.02, 110.57) | 520.84 | 97.75 (97.05, 98.46) | 434.76 | 9.58 (8.67, 10.49) |
| Sensitivity analysis 4 | 725.25 | −9.07 (−9.67, −8.47) | 702.97 | −30.64 (−31.18, −30.11) | 632.20 | −98.44 (−99.29, −97.58) | 512.28 | −215.81 (−216.87, −214.76) |

Notes:
[1] Average shingles cumulative incidence with chickenpox vaccination per 100,000 person-years (averaged over 30 or more model runs).
[2] Change in shingles cumulative incidence per 100,000 person-years calculated as the average shingles incidence with chickenpox vaccination minus the average shingles incidence without chickenpox vaccination. Positive number represents an increase in cumulative incidence and negative number—a decrease.
[3] Sensitivity Analysis 1—One-dose vaccination schedule; Sensitivity Analysis 2—Lower coverage rates; Sensitivity Analysis 3—Higher coverage rates; Sensitivity Analysis 4—Removing biological effect of boosting.
[4] Using the Mann–Whitney U-test all changes in cumulative incidence for every time and scenario combination were statistically significant ($p < 0.05$).

Our results illustrated that the short-term increase in shingles cases following chickenpox vaccination is largely dependent on the DoB and WoI—quantities whose values are still widely debated in the literature. Varying these values had a major impact on outcomes of chickenpox vaccination, with the percentage of increase in incidence rate ranging between 4.8% and 41.9% (25 years post-vaccination) between the most and least conservative DoB estimates. The only other study to vary the natural DoB, to our knowledge, was by *van Hoek et al. (2011)*, who found a short natural boosting and a longer shingles vaccine protection leading to variable increase in shingles following chickenpox vaccination. However, in this study, the DoB assumed one of only three values (7.5, 20, 42 years), only varied DoB in conjunction with shingle vaccine boosting, used a compartmental model and did not describe how they calibrated the model to empirical data (*van Hoek et al., 2011*). Our experiments with lower durations of boosting predicted increases in shingles cases post chickenpox vaccination smaller than previous models (*Brisson et al., 2010*, *2001*; *Marziano et al., 2015*). This is likely because, these DoB were significantly lower than previous models but are consistent with immunological assays used to measure the DoB in grandparents as described in the study above (*Ogunjimi et al., 2017*). Furthermore, the rate of boosting in our model was driven within an age- and distance-based transmission network rather than a random-mixing network, potentially limiting contacts that could produce a boost, which is supported by the fact that our average number of boosts per person was low at 1.83.
Our model results were congruent with other models that demonstrated, over a longer time horizon, shingles cases would decrease significantly following chickenpox vaccination. This decrease is because chickenpox vaccination decreases the burden of illness of chickenpox and the incidence of infection by VZV, thus reducing the population with dormant VZV, and producing a cohort effect as a greater and greater percentage of the population is vaccinated. Empiric studies post chickenpox vaccination have started to show evidence of this cohort effect, with younger age groups who have received chickenpox vaccination having lower rates of shingles than the corresponding age group prior to vaccination (*Marra, Chong & Najafzadeh, 2016*; *Humes et al., 2015*). The ultimate drop in shingles cases will depend largely on what percentage of chickenpox vaccinated cohorts are susceptible to shingles.

To date, empirical findings drawn from the era following introduction of chickenpox vaccination are largely inconsistent, making it difficult for policy-makers to know the continued impact of the chickenpox vaccine. While some studies show an increase in rates of shingles following the implementation of chickenpox vaccination, similar studies have shown that this increase started prior to vaccination, and other studies have found rates have stayed the same following vaccination (*Russell et al., 2014*; *Marra, Chong & Najafzadeh, 2016*). Different contact and boosting patterns (e.g., number of boosting events, age at which boosts occur, age of chickenpox infection) in different countries may also shape some of the observed differences in the age-specific incidence rates of shingles by country and potentially the impact of chickenpox vaccination by country (*Poletti et al., 2013*). A total of two Canadian studies argued that the incidence rate of shingles was increasing prior to chickenpox vaccination implementation and has stayed consistent following the implementation; however, such studies took place only seven and eight years following vaccination, and one study did not adjust for age (*Russell et al., 2014*; *Marra, Chong & Najafzadeh, 2016*). Our model demonstrated that at a lower DoB and higher WoI, the perceived impact on shingles rates would be quite small, increasing from 383 cases per 100,000 person-years to 406 per 100,000 person-years 10 years after chickenpox vaccination. This small increase may be difficult to measure or observe in empirical data where other factors influence the shingles rates (e.g., shingles vaccination, co-morbid infections, ageing of the population).

Our model produced some interesting secondary findings and observations. As with previous models, we found it challenging to account for the rate of shingles infection seen in the youngest age group (*Ogunjimi et al., 2015*). *Russell et al. (2014)* show a small but sudden increase in the rate of shingles infection in one- to four-year-olds. We theorized that a substantial proportion of these cases could be due to immunocompromising conditions in young children that may place them at a greater risk of developing shingles. However, it would be interesting to further explore if these individuals alone could account for this increase and if this is a trend that is found across countries. Furthermore, we found that varying chickenpox vaccine coverage (by changing vaccine attitudes within 10%) had only a minor impact on shingles incidence. However, greater changes to vaccine coverage levels would eventually realize larger impacts on shingles incidence. Evaluation of disease and vaccination outcomes stratified

by urban and rural populations demonstrated that urbanicity did not impact the overall results, suggesting that our model may be robust to uncertainty in population demographics within a given connection frequency between urban and rural population.

Although our model is one of the most detailed extant representations of the interaction between chickenpox and shingles, it is subject to limitations. First, we ran our model on a population of 500,000, raising the question of applicability to larger populations. However, we observed very little deviation in our findings when we ran the model on 50,000 vs. 500,000, suggesting a robustness of results to broad ranges in population size. Second, following a review of the literature, we decided not to include endogenous boosting, as the relevance of endogenous boosting is debated in the literature and *Ogunjimi et al. (2015)* found it insignificant. Third, we had difficulty accounting for shingles in younger age groups, and therefore had to adapt our model to fit Canadian data. Fourth, we were only able to vary a couple of the parameters relevant to the boosting of immunity; future research should further explore the impact of changing multiple boosting parameters (e.g., probability of boosting on contact by age). Finally, we used a stylised distance-based network to represent transmission of infection and did not implement a truly age-dependant contact matrix.

Future research and agent-based modeling should focus on studying some of the remaining unknowns surrounding the mechanisms of VZV reactivation, waning and boosting of immunity. ABMs could explore how changing the contact patterns alters the number and type of boosting events, and how this variation may explain the differences in shingles incidence both before and after chickenpox vaccination in different countries. Moreover, there should be ongoing comparison of model results and empirical findings post chickenpox vaccination, so we can update model parameters to fit with changing data. Our findings highlight the importance of not only studying when and if boosting occurs, but also the level of protection it confers on the individuals. While *Ogunjimi et al. (2017)* provides a good start to measuring the quantifiable impacts of boosting, studies should measure the longer-term immunological impacts of re-exposure and how those measurements translate to risk of reactivation. Furthermore, future research and models may want to look at the impact of other disease and population factors on the changing epidemiology of shingles, including immunocompromising conditions, co-morbid infections (e.g., CMV), and stress.

## CONCLUSION

Our model highlights the importance of not simply knowing when and if the VZV boosting events occur but the specific DoB, as these values can impact the effect of chickenpox vaccination on shingles incidence over time. Our study suggests that over the longer time period, there will be a reduction in shingles incidence driven mostly by the depletion of source of shingles reactivation, suggesting that in the long-term a universal chickenpox vaccine would be a good policy to reduce both chickenpox and shingles cases. However, in the short to medium term some age cohorts may experience an increase in shingles incidence. Our model offers a platform to further explore the relationship between chickenpox and shingles, including analyzing the impact of different chickenpox vaccination schedules and cost-effectiveness studies.

## ACKNOWLEDGEMENTS

Authors acknowledge Sowgat Ibne Mahmud and Nazifa Khan for their contributions to the conception and initial design of the agent-based model. Authors acknowledge contributions of public health department in Alberta for obtaining surveillance data and staff at the University of Saskatchewan Computational Epidemiology and Public Health Informatics Laboratory for use of their equipment to run simulation experiments.

### Funding

This study was funded by the grant from Alberta Health, University of Alberta project number RES0025110. The funders had no role in study design, data collection and analysis, decision to publish, or preparation of the manuscript.

### Grant Disclosures

The following grant information was disclosed by the authors:
Alberta Health, University of Alberta project number: RES0025110.

### Competing Interests

The authors declare that they have no competing interests.

### Author Contributions

- Ellen Rafferty conceived and designed the experiments, performed the experiments, analyzed the data, prepared figures and/or tables, authored or reviewed drafts of the paper, approved the final draft.
- Wade McDonald conceived and designed the experiments, performed the experiments, analyzed the data, contributed reagents/materials/analysis tools, prepared figures and/or tables, authored or reviewed drafts of the paper, approved the final draft.
- Weicheng Qian conceived and designed the experiments, analyzed the data, contributed reagents/materials/analysis tools, authored or reviewed drafts of the paper, approved the final draft.
- Nathaniel D. Osgood conceived and designed the experiments, analyzed the data, contributed reagents/materials/analysis tools, authored or reviewed drafts of the paper, approved the final draft.
- Alexander Doroshenko conceived and designed the experiments, analyzed the data, authored or reviewed drafts of the paper, approved the final draft.

### Ethics

The following information was supplied relating to ethical approvals (i.e., approving body and any reference numbers):

This study was approved by the Health Ethics Research Board at the University of Alberta, study ID Pro00068334.

## Data Availability

    1. McDonald, W., Rafferty, E., Qian, W., Osgood, N.D., Doroshenko, A. Chickenpox and shingles ABM. https://figshare.com/articles/Chickenpox_and_shingles_ABM/5294647/1.

    2. Rafferty, E., McDonald, W., Qian, W., Osgood, N.D., Doroshenko, A. Chickenpox and shingles ABM_connection and contact frequency profile. https://figshare.com/articles/Chickenpox_and_shingles_ABM_connection_and_contact_frequency_profile/5532763/1.

    3. Rafferty, E., McDonald, W., Qian, W., Osgood, N.D., Doroshenko, A. Sums of residuals squared used to determine scenario selection in model calibration and experimentation. https://figshare.com/articles/Sum_of_residuals_squared_-_VZV_model/5466556/1.

## Supplemental Information

Supplemental information for this article can be found online at http://dx.doi.org/10.7717/peerj.5012#supplemental-information.

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
