# Peer review of "Evaluation of the effect of chickenpox vaccination on shingles epidemiology using agent-based modeling"

_PeerJ, doi:10.7717/peerj.5012_

## Round 0.1 · original submission · Major Revisions

You have comments from three expert reviewers, of which two are anonymous and one is identified. Please address these comments in your revisions and response document.

One of the referees suggests changing the title of the paper. On this point, please do as you like; I won't insist on this change.

A note about code. One of the .docx files you uploaded crashes my installation of LibreOffice. Please provide something more user-friendly. Also the FigShare repo you provide seems to have the source as a PDF file, only (?). Is that right? I am a stickler for making source code available in usable formats, so please sort all this stuff out in your revisions. Before this paper will get sent out to referees, I will verify that the source code is provided in a user-friendly format. Note that I will insist on this.

Reviewer 1 ·

Basic reporting

See below.

Experimental design

See below.

Validity of the findings

See below.

Additional comments

In the current study Rafferty et al. present an agent based model of VZV transmission with an aim to evaluate the incidence of shingles after implementation of VZV vaccination campaigns. They calibrate the model to observed data in Alberta and find a temporary increase in incidence in certain age groups. The manuscript is generally well written and the modeling seems sound – though some details need to be better explained and some of the presentations reworked.

Comments:

- The study is set in Alberta; the title needs to reflect this. Something like: “Evaluation of the effect of chickenpox vaccination on shingles epidemiology in Alberta, Canada using agent-based modeling”

- Lines 133-136: all of the reasons you list for using ABMs apply to compartmental/ODE models as well. Better justification for ABMS is needed.

- AnyLogic is not familiar to the modeling community. A better explanation of the software is needed. The authors should also consider making the source available for replicability’s sake.

- Lines 216-219: the authors need to be more specific on the contact parameterizing. Why were 1-9 year olds chosen? Why not 1-5 or each year specifically? They also need to present the actual rates, perhaps in a contact matrix.

- The authors should be more explicit in their policy recommendations. Do they think deploying a vaccination program is a good idea?

- Figure 4 is confusing. Why is positive cases averted and negative cases added? It makes more sense to have up be more cases and down be fewer cases.

Minor comments:

- Ref 14 is incomplete.

Reviewer 2 ·

Basic reporting

'no comment'

Experimental design

'no comment'

Validity of the findings

'no comment'

Additional comments

• Reference No. 7 incomplete, therefore difficult to find.
• Reference 26, Idem. If the citation format from the journal does not allow to have access to the name of the chapter it will be difficult to check the reference source.
• In figure 1 I would separate each part of the figure (A, B, C, D) in boxes, as it gets confusing to read/interpret them without the text.
• I was not able to download the “Data reference 2” to better understand the scenarios or baseline scenario that were/was used.
• Figure 3 is missing the legend to differentiate what red and blue mean.
• Overall I think there is a problem with the treatment of “scenarios” information. Through my reading I had to keep going back and forth to learn the specific characteristics of each scenario. When I tried to understand the differences I referred myself to Table 2, however when explaining Table 3 in the results section (line 326), the authors talk about vaccination and non-vaccination scenarios, and at that point I felt that the reader is left out of the loop because from the list of ten scenarios presented in Table 2, there is not information about which ones are vaccination and non-vaccination scenarios. Hence, the specific details of each scenario should be stated.

·

Basic reporting

no comment

Experimental design

As it stands, the method is not described with sufficient information to be reproducible by another investigator.

Validity of the findings

no comment

Additional comments

This an interesting and well written paper about VZV epidemiology that suffers from a lack of description of the model assumptions and behaviours. Some sections (introduction) are particularly clear, and even if I have tons of questions and that the model is not described enough, I am very confident that it could be turned into a useful paper. One of the study strength is also that it is based on reliable demographic and epidemic data, and represent realistic assumptions regarding population vaccination attitudes.


1. The section describing the calibration and validation of the model was really unclear to me. The authors wrote that the model was calibrated to “the age-specific incidence of shingles and chickenpox prior to vaccination using a step-wise approach”, but the first step “calibrate our model output to age-specific incidence of chickenpox and shingles prior to vaccination”, which is the same quantity. Also, we don’t know how (method) this initial calibration was performed. I don’t understand how the chickenpox incidence can be fitted if only the shingles connection range is varied. Was there no uncertainty in the number of contacts and the risk of transmission on such a contact (mentioned line 186)?
Then, why was a second step needed? Is it because you wanted to inform the model calibration using another data, in the vaccine-era? You could have used this data in the first procedure and estimate your parameters / calibrate the model in one go. As it stands it looks like DoB and WoI have been estimated twice. Later in the manuscript, I found out that the particular parameter DoB wasn’t initially informed by data, as it was varying ~uniformly from 0.42 to 10 (steps of 1 between 2 and 10). Once again, it looks like it would have been simpler to vary this duration from 0.42 to 10 within a Latin Hypercube sample, and try to fit the model directly, just like you did with the other quantities.
It is interesting that the model could not fit the empirical age-specific shingles incidence rates when very short and very long duration of boosting were assumed. Perhaps a simpler calibration process could allow a proper estimation of realistic values of DoB and WoI.


2. Line 214: the authors state that their model represent a low-density and a high density region. Adding this “dimension” into the analysis is extremely pertinent, but there is absolutely no other information or result that mention this model feature. For example, we have no idea what is the relative size of each region.
VZV hazard is one of the most remarkable marker of human connectivity. As an example, in 1928 Fales showed that children residing in rural Maryland were infected 2 years after those residing in urban Maryland (Fales WT (1928) The age distribution of whooping cough, measles, chicken pox, scarlet fever and diphtheria in various areas in the United States. Am J Hyg 8: 759–799). It has already been shown that ABM’s have the capacity to reproduce the current socio-spatial heterogeneity in VZV hazard (Silhol R, Boëlle P-Y (2011) Modelling the Effects of Population Structure on Childhood Disease: The Case of Varicella. PLoS Comput Biol 7(7)), so literature suggests that this model feature could give useful results.
The paper would really benefit from a better description of the differences between the two areas, and show results stratified by region density (e.g. mean age at varicella infection when the vaccination function is disabled, distribution of the number of boosts in each area). Did the authors assume that the chickenpox and shingles incidence in the pre-vaccine era were the same across all regions? Is there data that could support this assumption? This should be discussed.


3. Line 212: Please include some references on the distance-based network. The reader needs to know what is this “proximity” and what the contact rates are. Would it be possible to describe and show the resulting intensity of contacts between the different age groups?
The authors say that “An agent’s chance of being infected with chickenpox was dependent on whether they came into contact with someone with infection and the risk of transmission on such a contact”, but we don’t know how these two quantities were parametrised and used by the model.

4. According to CDC, the majority of breakthrough chickenpox cases are 1/3 as infectious as those of unvaccinated individuals (https://www.cdc.gov/chickenpox/hcp/clinical-overview.html). It would be useful to know what is currently assumed by the model, and how does it impact its predictions on shingles incidence.

5. From the line 132 to 142, it seems like the authors make a series of statements that implicitly compare ABM’s and compartmental models. I would really suggest to be more precise and cautious when making these types of statements. For example “simulate the indirect effects of varicella vaccination, including the changing risk of disease over time, herd immunity and increasing age of infection”, can be done by an age-structured compartmental model. I would really suggest this section to be reviewed by someone with a strong background in both ABM and compartmental models.

6. Line 165: It would be useful to know why was a burn-in period needed.

7. The cumulative incidence (or seroprevalence) of VZV by age before vaccination is introduced is now very standard (see this recent paper: Marangi L, Mirinaviciute G, Flem E, Scalia Tomba G, Guzzetta G, Freiesleben de Blasio B, et al. (2017) The natural history of varicella zoster virus infection in Norway: Further insights on exogenous boosting and progressive immunity to herpes zoster. PLoS ONE 12(5): e0176845.), and can be easily outputted by the model. Please state why the model wasn’t fitted on this quantity (data gap in the pre-vaccine era?).

8. Some model parameters are not described and sourced, such as the mortality rate, which could impact the results since ageing is linked with shingles incidence.

9. From what I understood, at least runs 30 were performed for one single parameter set (scenario). The text could mention the fact that this high number of agents should almost remove any stochasticity (judging from your figures).

10. Please define exogenous infection rate (line 255). Is it related to shingles only, as suggested by your introduction?

11. It would be useful to give the reader more information about the distribution of the age at first shingles case. As the authors mention earlier, a vaccine against shingles is available and this distribution could inform the age at which individuals should get vaccinated.

12. I also found that some terms could be used more consistently throughout the manuscript
Line 92, perhaps consider replacing “population” by “persons” ?
Line 118 and 264, replace “case” or “rate” by “incidence”
Line 315: “tried to fit model to empirical data”. Same with Figure 2


Table 1: some units are missing. For example, is the force of reactivation a yearly rate? How is the exogenous infection rate actually used in the model? We also don’t know what “preferential” and “normal” distance mean, how it is used etc… . It would have been useful to know which prior range was tested during the calibration process, as we currently only have the posterior values. What do “preferential” and “ordinary” contacts mean? I don’t think it is defined or mentioned anywhere.

Figure 2: Please state what the blue polygon represent (pointwise minimum and maximum values?).

Table 3: First column. Is the “cumulative incidence” the mean incidence of shingles per year calculated over the first 10 years following vaccination, averaged over the 30 or more runs?

Table 4 line 5: perhaps replace “was calculated” by “, calculated”

Supplemental figure 2 and 3: I think it would be clearer to say as a first sentence that what you showing are simulated and empirical age-specific shingles incidence rates. Once again, we also need to know if the blue polygons represent pointwise min/max or 95% uncertainty interval, or something else.

Also, there was a problem when I tried to download the file “Sums of residuals squared_VZV_ABM.csv” ({"message": "Entity not found: file", "code": "EntityNotFound"})

---

## Round 0.2 · Major Revisions

You have reviews from two of the three original referees.

Referee number three, who has chosen to make his review public (i.e., non-anonymous), still has concerns about the clarity of the presentation of the methods.

Given that these concerns pertain to explaining your approach more transparently — and not the scientific validity of what you have found — it should be straight-forward to address them.

Reviewer 1 ·

Basic reporting

no comment

Experimental design

no comment

Validity of the findings

I still disagree that the title accurately reflects what is presented in the manuscript. As the authors state in response to Reviewer 3’s point #2: “…our aim in including the low and high-density regions was not to compare the results in the two groups but to ensure an accurate representation of a typical public health region in Alberta (or in Canada or other comparable jurisdiction)…”. I still suggest changing the title to something like “Evaluation of the effect of chickenpox vaccination on shingles epidemiology in Canada using agent-based modeling”.

Additional comments

Rafferty et al. have revised their manuscripts answering my comments.

·

Basic reporting

no comment

Experimental design

The method and model behaviours are still not described with sufficient information to be reproducible by another investigator.

Validity of the findings

no comment

Additional comments

I think that the authors answered satisfactorily to some of my comments, and that the method is better described overall, but large parts of the revision are unsatisfactory as the model behaviours are not sufficiently described.


As raised previously by reviewer 1 and I, we need to know the total number of contacts (you can call it contact rate) that occur between the different age groups (perhaps over the first year or first month of the simulation).

I am a little bit confused by the way the probabilities of infection per contact were parametrised, and I think this needs clarification. An agent makes a certain number of contact every day and three parameters describe the probability of the three types of infections during a particular contact. The probability of “normal” chickenpox infection for such single contact is 0.78 (78%) in the model, whereas in the Ceyhan et al. (2009) study, which is cited as the parameter value source, 78% is a probability of infection (the SAR is a good proxy for it in this particular case) calculated over (roughly) 21 days of contacts with an infectious individual in the household. Please include one or two sentences in the method to explain how these probabilities were derived. It is quite important since these three values are fixed throughout the analysis, and as such, considered as “exactly known”, which is quite uncommon for transmission probabilities.

From what I understand, there is no input or output information which is stratified by “urbanicity”, except a number of contacts, but we still don’t know how this heterogeneity is parametrised. Basically, we don’t know how, in practice, these numbers of contacts differ between places. If “rural” individual have less agents within their connection range than “urban” ones, then I understand that the connection range can be assumed to be the same for urban or rural individuals, but another parameter has to vary. It seems to me that the table 1 doesn’t present such parameter.

I don’t understand why we can’t have a sense of the incidence of chickenpox or shingles in both rural and urban areas, as it is an important check of the contact and transmission assumptions, and provides useful information. I have to say that this omission really contrasts with the section that describes how detailed can be an information when provided by an ABM. As mentioned earlier, there seems to be no empirical input or fitting data describing rural/urban differences, so this population stratification should be justified, the differences between the settings (e.g. number of contacts per individual per week/months/year) should be clearly measured, and the differences in epidemic outcomes should be evaluated. If there are no differences in incidence or vaccination impact by urbanicity then one line in the discussion will be enough. it could be that urbanicity doesn’t influence the results, making them more robust to uncertainty in population demograpics and assumptions about the number of contacts, but the reviewers and readers need to be reassured that every possible model outcomes was calculated and checked against data or common sense (=the downside of complex models!).

The proportion of VZV-naïve individuals by age in the model (pre-vaccine era) should be at least be visualised as a model check, as it is related to the first stage of your fitting process, and it will impact your model results. There is good data on the population of Newfoundland (S Ratnam. Varicella susceptibility in a Canadian population. Can J Infect Dis 2000;11(5):249-253. ), and I think their proportions could be compared to your model outcomes, even if Alberta is more densely populated. Once again, I think that every model should be fully checked before being used for publication, especially if they are relevant to public health issues. If the model disagrees with the data, then this needs to be discussed.

I have been able to download and visualise the “contact frequency profile” under the form of a heatmap, but I don’t know how this matrix has been computed. The sum of each column is 1 which is a little bit confusing because the matrix is triangular. Please include more details about how this matrix was calculated.

Please mention at line 179 that one day is the model time step (if it’s the case).

Line 184-188, I think you are describing probabilities and not “infection rates”.

It I understood well, the model is first fitted on age-specific incidence of shingles and chickenpox prior to vaccination, but chickenpox incidence data is re-scaled by the model during the same step of the fitting process. I think this should at least be mentioned in the discussion.

Line 222, please explain what the “AnyLogic calibration experiment” is. Does it maximise the model likelihood? It is not a standard method.

Is there a reason why you don’t use the actual years throughout the paper and don’t assume a certain year for the start of vaccination (could be 2002)? It is not mandatory but I think showing results with the actual years would better illustrate your results (e.g. figure 3 with “2002” instead of “0”, etc…). Similarly, “2002-2012” would be more explicit that “T10” on your figure 4, and it seems to me that in your case, it would be fair to assume that vaccination started in 2002.

Figure 1, the four flowcharts are very nice but it looks like what is shown are not state names but state variables names as they were coded. Even if it’s not very nice (I am not asking this to be changed), the coded names are explicit enough, but could you mention somewhere what “PHN” is? The flowchart “A” could be a little less confusing if a little text at the right side of “shingleVacc” was added to state that vaccination against shingles was not modelled (already mentioned lines 142-145).

Please add a table showing which data that was used for calibration/fitting: outcomes (with age groups), years of data collection, source. At the moment this information, especially the years of data collection, are not provided.

Some figures have misplaced legends (hiding the data), such as Figure 2 (or Figure S4).

Figure 4: I think you should be consistent with age ranges and present values until 100 years old.

Table 1 should show the values of the (key) model parameters. Please replace “Range of values tested” by both prior and posterior parameter distributions so that we will know what was tested and how data informed the model.

Figure 2 and S2-4: Please specify which year (or when, e.g. “before chickenpox vaccination is introduced”) this has been calculated. I guess it is “time 0”, but it is not clear.

---

## Round 0.3 · Minor Revisions

This is moving in the right direction but the reviewer still has some reasonable questions. I look forward to your re-submission which will address them. The reviewer notes (and I agree) that not all models need be the same to be good contributions. However (and here I also agree) that, at the least, some discussion is needed of why it's profitable (and/or reasonable) in this instance to use differing probabilities of chickenpox infection per contact than have been used before. Et cetera, with his other remarks.

What is more, some of the smaller comments are objectively correct (for example, Suppl. figs 3A/D lack a caption). Thank you.

·

Basic reporting

no comment

Experimental design

no comment

Validity of the findings

no comment

Additional comments

1. There are consistent estimates of the probabilities of chickenpox infection per contact or day, deriving from models using time-use data (Zagheni, Am J Epidemiol 2008, ~100 citations), social contact matrices (Ogunjimi, already in your list of references), Census data (Silhol, PloSCompBiol 2011) etc…. and they are ~5 times lower than the probability you assume to be exactly known in your analysis (16% vs 78%). You end-up having a secondary attack rate (in your model, calculated as the proportion of contacts that become infected by a single newly-infected index case over the course of the epidemic?) that is twice lower than your probability of infection per one contact/day, which is paradoxical. By the way, line 240 “We considered an attack rate in our model (41%)…” is unclear, as we don’t know if it is an assumption, a model outcome, and how it is calculated.
I think it is a strength for our field to have different models using different assumptions, “distance-based network” vs “network structured by household, communities etc…”, but your assumptions about the transmissibility of VZV need to be seriously discussed because they hardly make sense when compared to other estimates and common sense. I think most of the differences could be simply due to your network that is less “structured” than in other models (despite accurately representing age-mixing). In short, using an SAR (originally calculated within households over 2-3 weeks) as a fixed probability of infection per contact/day is clearly problematic here and should raise a lot of questions as transmission probabilities are the main driver of VZV incidence. A more appropriate way to handle the uncertainty in these probabilities of transmission per contact could be to acknowledge it and use a range based on published estimates, and fit these probabilities to the data you have.

more minor point:
2. Looking at your Supplementary Figure 3A/D (by the way, the figure caption is missing). The chickenpox incidence levels are the same in urban/rural settings (looks like a saturation effect where a “low” density is dense enough to produce the same epidemic levels as a “high” density area), but the incidence curves are overlapping so much that it looks as if there was a lot of contacts between the two areas. What was your assumption about that? For example, what is the proportion of contacts of rural agents that belong to the urban area?
The shingles incidence looks slightly higher in rural settings is the absence of vaccination (could due to lower number of boosts?), but becomes lower when there is vaccination (I find difficult to interpret this!!). I think it would be interesting to mention these differences, as it seems that you would like different jurisdictions (more or less dense) to use your result when considering implementing chickenpox vaccine. It could also be considered as a sensitivity analysis.

---

## Round 0.4 · Minor Revisions

Your revised manuscript is not clear to a subject-matter expert, so I am returning it to you for further revisions.

I do not like to have so many rounds of revisions any more than you do. However, if you put yourself in my shoes, I have a duty to make sure that the journal's articles are clear. The referee, who is a subject matter expert, is not clear on your approach, which says to me that other readers will not be clear, either. I must confess that I am also having trouble figuring out where the 78% comes from.

I would like you to take one more pass at it. Please pay especial attention to clear explanation on how your model works, what the parameters mean, and where they come from. "Calibration" is not an explanation. If you need more time you can have as much as you like; the computer-generated deadlines are not hard deadlines.

I cannot recommend publication as long as the referee feels its unclear. If we are still at an impasse after another round, I will be forced to seek a new reviewer, which (presumably) nobody wants.

Thank you.

·

Basic reporting

No comment.

Experimental design

No comment.

Validity of the findings

No comment.

Additional comments

1) I am really sorry, as it could only me that has this problem, but I still don't understand how VZV spreads within the model, and the terms that you are using seems to misled me. It should be noted that you have made an effort to better explain it, and I think that in the example line 253 is particularly useful. As I said during the first review, the study and the results still make sense to me, but...

Here is more precisely what I don’t understand and why I think the model should be presented in a clearer way:
"78% (as shown in Table 1) is the probability of infection per contact (message) received by a susceptible agent." (from a normal chickenpox-infected agent, and 23% if the agent has breakthrough infection or shingles)
Then,
"An infected agent in the preferred age group (i.e. a child) may send 20 messages per day, each to a randomly connected agent. Considering that the mean number of connections per agent in this age group is 10.87, this gives us about 20 / 10.87 = 1.8 ≈ 1 to 2 messages per connection-day. "
So, from what I read, if one susceptible child and one infectious child are connected together, they usually have 1 or 2 contacts between them, and the probability of infection over one day is 78% if they had one contact during the day, and something like 95% if they had two contacts.
For most of the readers, the term “probability of infection on contact” will relate to the first principles of epidemic modelling, such as the R0 definition. I don’t know how this 78% probability should be named if the “true” probability of infection is actually derived from this original 78% probability, as well as from network characteristics at that time point. Since your ABM explicitly modelled the contacts between the susceptible child S and the infectious child I, and since you also know the probability of infection between S and I, how can other quantities come into play when deciding if S is going to be infected? Perhaps 78% is not a “probability of infection on contact”, but something else. Perhaps a diagram would help, I don't know...
And, as a result, in the table 1, surely this 78% probability can’t be fixed and at the same time “calibrated”. By the way, I don’t think that we know what the durations of the VZV infectious stages are.


2) It is something we might have debated quite a lot already (sorry!), but I still disagree with the authors about understanding the model behaviours when both population density and number of connections are decreased. Since at the same time you assume a similar number of contacts and a same transmission probability per contact within “rural” and “urban” areas, it seems to me that it would be important to understand why you obtain the same chickenpox epidemic before vaccination.


Romain

---

## Round 0.5 · accepted · Accept

As I said in our previous exchange, if you you could not satisfy the reviewer requesting more changes, I would go to a new reviewer. This is what happened.

I can report that the new reviewer recommends acceptance of the paper, and that, therefore, I recommend the same to the journal. Congratulations.

·

Basic reporting

No comment

Experimental design

No comment

Validity of the findings

No comment

Additional comments

Despite the fact that the model calibration is now better described, I am really sorry but I still consider that the model hasn’t been fully checked and should not be published as it is. My questions regarding its strange behaviours have not been answered despite a few iterations and it looks like my questions will remain unanswered, so I think the editor should do without me.

The fact that the chickenpox incidence rate (without vaccination) is the same when the parameter modelling the number of agents per distance is decreased by 33% should be explained, otherwise it will really look like you only partially understand what is happening in the model. By the way, I am not sure that including population density in the model design is really compatible with the pattern-oriented modeling approach since you don’t have any empirical patterns/data of VZV/HZ incidence by regions/density.

The duration of the chickenpox infectious period is still not reported.

Finally, if an index case sends 20 or 23 messages per day to 10 or 17 connections (let’s assume here that 5 of them are susceptible), and if the resulting varicella SAR is 40% in the model (which is totally fine), 2 out of these 5 susceptible individuals got infected during the two weeks of varicella infectious period (on average). Something like 300 or 350 messages would have been sent by the index case during the 2 weeks, and it seems unlikely that only 2 infections among 5 susceptible would have happened if the probability of infection for one single message is 78%, even if the pool of contacts comprises 20/30 individuals.

Romain

·

Basic reporting

This article is generally well written. It includes sufficient details for readers to understand the model and interpret the simulation results. The figures and tables are well designed and presented.

Experimental design

The study has a sound experimental design.

Validity of the findings

The authors provided sufficient details for others to replicate the model and results. Although the model includes several simplifying assumptions, which are necessary for any simulation models, the authors have sufficiently justified the use of these assumptions.